# Polymer Microspheres and Their Application in Cancer Diagnosis and Treatment

**DOI:** 10.3390/ijms25126556

**Published:** 2024-06-14

**Authors:** Mingyue Zhai, Pan Wu, Yuan Liao, Liangliang Wu, Yongxiang Zhao

**Affiliations:** State Key Laboratory of Targeting Oncology, National Center for International Research of Bio-Targeting Theranostics, Guangxi Key Laboratory of Bio-Targeting Theranostics, Collaborative Innovation Center for Targeting Tumor Diagnosis and Therapy, Guangxi Medical University, Nanning 530021, China; zhaimingyuewasd@163.com (M.Z.); wupan@gxmu.edu.cn (P.W.); liaoyuan2024@163.com (Y.L.); wuliangliang1211@126.com (L.W.)

**Keywords:** polymer microspheres, cancer, diagnosis, treatment, repair

## Abstract

Cancer is a significant global public health issue with increasing morbidity and mortality rates. To address this challenge, novel drug carriers such as nano-materials, liposomes, hydrogels, fibers, and microspheres have been extensively researched and utilized in oncology. Among them, polymer microspheres are gaining popularity due to their ease of preparation, excellent performance, biocompatibility, and drug-release capabilities. This paper categorizes commonly used materials for polymer microsphere preparation, summarizes various preparation methods (emulsification, phase separation, spray drying, electrospray, microfluidics, and membrane emulsification), and reviews the applications of polymer microspheres in cancer diagnosis, therapy, and postoperative care. The current status and future development directions of polymer microspheres in cancer treatment are analyzed, highlighting their importance and potential for improving patient outcomes.

## 1. Introduction

Cancer has emerged as a significant global public health concern, with increasing prevalence and mortality rates straining healthcare systems and resources [1]. While current cancer treatments such as surgery, chemotherapy, and radiotherapy have been somewhat effective in managing tumor growth and spread, they are hindered by various limitations and obstacles, leading to persistently high mortality rates. Surgical resection, the most direct approach, encounters challenges when dealing with cancers that have metastasized or when tiny undetected cancer cells are present [2]. Chemotherapy targets cancer cells with drugs, but resistance can develop over time, diminishing its effectiveness and causing harm to healthy cells [3]. Radiotherapy utilizes high-energy radiation to destroy cancer cells; however, there are limits to the dosage to avoid harming normal tissues. Additionally, some cancer cells within tumors may be resistant to radiotherapy, leading to potential recurrence [4]. Therefore, the quest for a new, highly effective cancer treatment that minimizes harm to the patient’s body and extends survival has become a shared aspiration among patients and researchers worldwide.

In recent years, the field of anticancer drug delivery has witnessed the emergence of several innovative approaches. Traditionally, anticancer drugs have been primarily administered through intravenous injection, a method that poses challenges in precisely controlling the speed and location of drug release [5]. Following administration, the drug disperses into the bloodstream across the body, leading to significant fluctuations in blood concentration. Consequently, patients are required to take the drug frequently and for extended periods to maintain therapeutic effectiveness. To address the issue of drug toxicity to healthy tissues and enhance therapeutic outcomes, researchers have introduced a novel drug delivery system (DDS) [6]. A DDS offers the advantages of enhancing drug bioavailability, minimizing side effects, regulating release rates, and extending drug circulation within the body [7]. This innovative system has garnered considerable attention within the medical community. Notably, polymer microspheres serve as a form of DDS. Polymer microspheres are minute spherical structures formed by dissolving or dispersing a drug within a polymeric material, with diameters typically ranging from 1 to 1000 microns [8]. Drugs can be encapsulated within microspheres or attached to their surface through physical or chemical interactions. Polymer microspheres offer various administration routes, including subcutaneous injection, intra-tumor injection, or inhalation as agents for pulmonary delivery. Furthermore, they provide protection to the drug against environmental factors that could lead to inactivation [9]. Based on the material used for their construction, polymer microspheres are categorized as biodegradable or non-biodegradable [10]. Biodegradable polymer microspheres, in particular, are preferred due to their excellent biocompatibility and degradability, making them a vital component of a DDS.

This review examines the recent advancements of polymer microspheres in cancer research, specifically their role in cancer therapy. The review begins by discussing the importance of selecting appropriate polymer microsphere materials as the initial step in their production. It then delves into various fabrication methods such as emulsification, phase separation, spray drying, electrospray, microfluidics, and membrane emulsification, providing a concise overview of each method’s characteristics. Subsequently, the review explores the applications of polymer microspheres in cancer diagnosis, treatment, and postoperative repair, highlighting their current usage in cancer therapy (Figure 1). Finally, the review evaluates the current status and challenges of polymer microspheres in cancer research and provides insights into their future development direction.

## 2. Polymer Microspheres Overview

Polymer microspheres are commonly used as drug carriers for slow-release applications. The performance of these microspheres depends not only on the drug they contain but also on the choice of polymer material and the preparation techniques employed. The ideal sustained-release microsphere formulation is a result of the combination of the active substance, preparation technology, and polymer material. This section will systematically describe the raw materials and preparation techniques used for microsphere preparation, aiming to offer comprehensive theoretical and practical guidance for enhancing the performance of polymer microspheres.

### 2.1. The Material for Polymer Microspheres

The nature of microsphere carrier materials categorizes them into two main groups: biodegradable and non-biodegradable [10]. Non-biodegradable microspheres can accumulate in the body post drug administration, leading to toxicity issues and complicating treatment. On the other hand, biodegradable microspheres automatically degrade in the body into harmless substances, eliminating the need for additional surgery to remove any of the remaining polymer matrix [11]. Consequently, the use of biodegradable materials in microsphere preparation is increasing in current research. Biodegradable microspheres can further be classified into natural polymer microspheres and synthetic polymer microspheres, based on the polymer source [12]. This classification aids in a more precise understanding of microsphere properties and applications, expanding the scope for research and development in the medical field.

Natural polymers, derived from natural sources, are known for their exceptional biocompatibility, low toxicity, and biodegradability [13]. Examples of these polymers include gelatin, alginate, chitosan, silk protein, collagen, and hyaluronic acid [14]. In comparison to synthetic polymers, natural polymers offer advantages such as intricate structures, excellent biocompatibility, degradability, and renewability [15]. The porosity, charge, and mechanical strength of natural polymers can be finely tuned by adjusting their concentration and polymerization conditions, as well as through the incorporation of functional groups. Furthermore, the addition of chemicals, proteins, peptides, and cells can enhance their biological activity [16]. Due to these unique properties, natural polymers have been extensively studied for regenerative medicine applications and are commonly used in drug-carrying microspheres for cancer therapy [17]. For instance, Yang et al. utilized sodium alginate to create drug-carrying microspheres [18], while Chen et al. and Wang et al. developed microspheres containing chemotherapeutic drugs using gelatin [19,20]. Similarly, Lee et al. and Lei et al. utilized hyaluronic acid and silk protein, respectively, to produce drug-loaded microspheres [5,21]. These examples underscore the broad utility of natural polymers in the realm of drug-loaded microspheres for cancer therapy. However, the properties and bioactivities of natural polymers are significantly affected by their sources and extraction methods. Hence, to effectively utilize natural polymers in microsphere production, it is crucial to standardize manufacturing processes [14]. Only through standardized procedures can we fully harness the advantages of natural polymers to offer more efficient and safer solutions, especially in fields such as cancer therapy.

Synthetic polymers are created through the polymerization of monomers and offer numerous advantages over natural polymers. They have a longer degradation cycle, strong mechanical properties, adjustable chemical characteristics, are easy to reproduce, and are cost-effective to manufacture [22]. The main types of synthetic polymers include polyesters, hydrogels, polyamides, and polyurethanes. Notable examples of synthetic polyesters include poly(ε-caprolactone) (PCL), poly(lactic acid) (PLA), and poly(ethanol-co-lactic acid) (PLGA), which are commonly used and FDA-approved for various biomedical applications [23]. Hydrogels such as poly(ethylene glycol) (PEG), poly(vinyl alcohol) (PVA), and poly(acrylamide) (PAM) are preferred in tissue engineering due to their resemblance to the extracellular matrix, which facilitates cell growth [24]. Additionally, poly(orthodontic ester) (POE) can effectively modify microsphere behavior in degradation and drug release by incorporating acidic or basic excipients [25]. Polycarbonate-based compounds are widely used in surgical sutures, drug carriers, and tissue engineering because of their biocompatibility and biodegradability [26]. In the practical application of synthetic polymers, various researchers have adopted unique approaches. Cao et al. successfully overcame the resistance of tumor sites to chemotherapeutic drugs and effectively suppressed the growth of osteosarcoma using hydrogel microspheres composed of collagenase and PLGA microspheres [27]. Xiong et al. developed porous PLGA microspheres loaded with natural drugs using the solvent evaporation method from an emulsion, introducing an innovative approach for lung cancer treatment through drug delivery [28]. Wang et al. utilized heparin-coupled PVA microspheres as adsorbents to selectively eliminate tumor-induced immunosuppressive cytokines [29]. These studies not only showcase the diverse applications of synthetic polymers in medicine but also introduce innovative ideas and techniques for future cancer therapy.

### 2.2. Preparation of Microspheres

Various methods are utilized for microsphere preparation, such as emulsification, phase separation, spray drying, electrospray, microfluidics, membrane emulsification, and others (Table 1). The choice of method depends on factors such as the physicochemical properties of the drug being encapsulated, the characteristics of the polymer material, and the specific requirements of the microspheres for their intended application.

#### 2.2.1. Emulsification

Emulsification is the primary method used in the preparation of microspheres. It can be categorized into single emulsification and double emulsification methods based on the variation in emulsification techniques. The selection of the appropriate emulsification method depends on the specific properties of the drugs and polymer materials involved, ensuring efficient drug encapsulation and release in the microspheres.

##### Single Emulsification

Single emulsification involves dispersing a liquid (typically a mixture of the drug and polymer) at high-speed shear into another immiscible liquid with an emulsifier to create an emulsion. The solvent is then removed, or physical/chemical methods are used to solidify the droplets, resulting in microspheres [30]. This method is ideal for encapsulating hydrophobic drugs. For instance, Lee et al. utilized a single emulsification-solvent evaporation technique to encapsulate the hydrophobic drug erlotinib in PLGA materials for sustained-release microspheres [5]. Additionally, Xiao and Wei et al. employed single emulsification-crosslinking curing to develop chitosan microspheres and magnetic microspheres, respectively [38,39]. While the single emulsification technique is straightforward to use and has a low equipment cost, the particle size distributions of the microspheres produced tend to be wider due to increased aggregation and deformation of droplets during emulsification.

##### Double Emulsification

Double emulsification typically involves initially dispersing one liquid into another immiscible liquid to create a primary emulsion, which can be of the water-in-oil (W/O) or oil-in-water (O/W) type. This primary emulsion is then further dispersed into a third immiscible liquid to form a more complex emulsion, such as W/O/W or O/W/O, leading to the production of microspheres through solvent removal or droplet solidification [31] (Figure 2a). For instance, Xiong et al. utilized the W1/O/W2 double emulsion method to fabricate porous PLGA microspheres containing artemisinin, with ammonium bicarbonate serving as a pore-forming agent [28]. This method allows for the encapsulation of both hydrophobic drugs and hydrophilic compounds, overcoming the limitations of single emulsification. While proteins can also be encapsulated using this technique [40], it does have drawbacks such as complexity, the need for precise equipment and techniques, and relatively low productivity. Despite these challenges, the multiple emulsion structures formed during double emulsification help to limit aggregation and deformation between droplets. This results in microspheres with a more uniform particle size and morphology compared to single emulsification. However, the size distribution of the microspheres still tends to be quite broad overall [31].

#### 2.2.2. Phase Separation

The phase separation method for preparing microspheres involves adding organic non-solvents or inorganic salts to reduce the solubility of the polymer in the solvent, leading to phase separation and the formation of drug-carrying droplets that solidify into microspheres [32] (Figure 2b). This technique relies on the physicochemical phenomenon of extraction, with various factors influencing microsphere properties, such as the intermolecular interactions of polymer materials and the speed and duration of each preparation step. Huang et al. demonstrated the creation of chitosan microspheres with a porous structure through low-temperature thermally induced phase separation for a three-dimensional cell culture [40]. Similarly, Wu et al. developed alginate microspheres as a hemostatic agent using this method [41]. The phase separation method offers advantages such as cost-effectiveness and batch processing ease, but drawbacks include potential microsphere aggregation, challenges in dispersion, high residual organic solvent levels, and difficulties in ensuring sterility.

#### 2.2.3. Spray Drying

Spray drying is a method used to dry and solidify liquid materials, such as polymers and pharmaceutical ingredients, into microspheres. This process involves dispersing the liquid into fine mist-like droplets using an atomizer, which then quickly evaporate upon contact with hot air at high temperatures [42,43] (Figure 2c). Wei et al. utilized starch as a raw material to produce starch hydrogel microspheres through the spray drying method [44]. One of the key advantages of this method is the rapid and continuous one-step generation of particles without the need for additional drying processes, enhancing production efficiency. Additionally, the absence of external solvents in the preparation process makes it suitable for a wide range of drugs, especially those containing biologically active proteins and peptides [33]. This not only reduces drug loss but also enhances encapsulation rates. However, drawbacks of the spray drying method include the adherence of undried raw materials to the inner walls of the instrument, leading to material loss, and the significant impact of drying temperature on microsphere properties. High temperatures can cause deformation and aggregation, while low temperatures may result in solvent residue and poor control of particle size. These challenges should be carefully considered during practical implementation [6]. 

**Figure 2 ijms-25-06556-f002:**
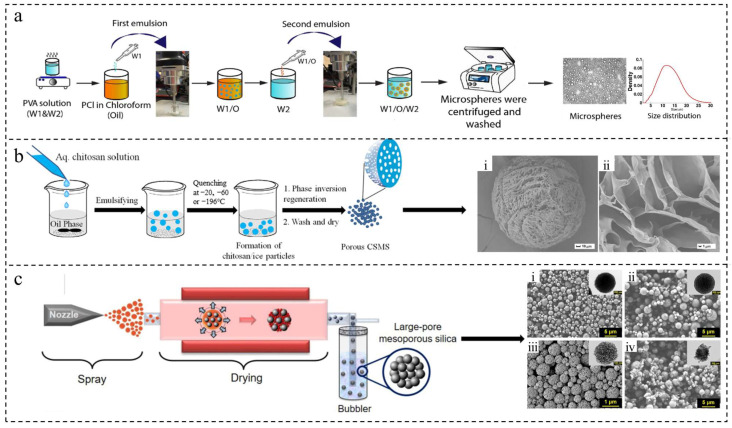
Microspheres were prepared through emulsification, phase separation, and spray drying. (**a**) The mechanism of preparing microspheres using the double emulsion method [31]. Copyright 2022, Elsevier B.V. (**b**) The mechanism of preparing microspheres by the phase separation method [32] and (i,ii) SEM image of the prepared microsphere. Copyright 2017, Elsevier B.V. (**c**) The mechanism of preparing microspheres by the spray drying method [43] and (i–iv) SEM and TEM images of the samples prepared by the salt-assisted aerosol-based method at salt-to-SiO_2_ molar ratios of 0.5, 1, 1.75, and 2. Copyright 2020, Elsevier B.V.

#### 2.2.4. Electrospray

Electrospray (ES), also known as the electrospray or electro-atomization method, involves a high-voltage power supply, syringe pump, nozzle, and collector. Liquid material is injected into the capillary nozzle using the syringe pump. Under the influence of a high-voltage electric field, the material passes through the nozzle to form a highly charged conical shape, known as a Taylor cone. The charged droplets are then dispersed to the collector due to the electric field force while the evaporation of solvents or a chemical reaction occurs during the injection process, ultimately resulting in solid microspheres [19] (Figure 3a). ES technology encompasses conventional ES, coaxial ES (CES), and triaxial ES (TES). In CES, the coaxial needle consists of inner and outer needles and is commonly used for creating drug-encapsulating microspheres. TES involves three needles—the outer, center, and inner needles—and is used for designing multilayer particles [45]. By adjusting various experimental parameters such as voltage and flow rate, the size and morphology of microspheres can be controlled to achieve microspheres with a controllable particle size and high encapsulation rate. Additionally, complex structures such as core-shell and porous structures can be obtained [34]. Chen et al. utilized ES to produce microspheres with various sizes and morphologies by adjusting the voltage, needle diameter, and flow rate [19]. ES has the capability to adjust particle size flexibly and achieve a high drug encapsulation rate. It can be used for a variety of liquid materials and to prepare microspheres with complex structures. However, this technology comes with drawbacks such as high equipment costs, complex operation, material property requirements, slow preparation processes resulting in low yield, and the use of highly volatile solvents that pose safety hazards and pollution issues.

#### 2.2.5. Microfluidics

Microfluidics involves manipulating fluids through micron-scale channels using external forces to control fluid behavior [46]. In the context of microsphere preparation, it typically involves injecting immiscible fluids into micrometer-scale channels to create droplets that can be further processed into microspheres [47] (Figure 3b). Common geometries used for microsphere preparation include T-shaped, co-flow, and flow-focused channels. Jiang et al. demonstrated the successful preparation of gelatin methacrylate microspheres using a one-step microfluidic technique [48]. The method allows for precise control of fluid flow rates, resulting in uniform and reproducible microspheres. Its advantages include high experimental control, high-throughput preparation [35], versatility in materials and reaction conditions, and the ability to produce various types of microspheres [36]. Utilizing micron-sized channels allows for the use of reagents in small quantities, leading to reduced waste and lowered costs. However, microfluidics also presents challenges. For instance, the equipment requires precision machining, which adds to the overall expense. The operation involves intricate processes that require precise control and micromachining techniques. Moreover, the size constraints of microfluidic channels limit the scalability of microsphere preparation, making it unsuitable for mass production. Lastly, maintaining the cleanliness of the equipment is essential due to costly and time-consuming maintenance requirements.

#### 2.2.6. Membrane Emulsification

Membrane emulsification involves creating an emulsion by applying pressure to drive the dispersed phase through micropores in an inorganic membrane into the continuous phase. As the dispersed phase passes through the membrane under pressure, the continuous phase flows over the membrane surface. Once droplets reach a specific size, they detach from the membrane and enter the continuous phase [37]. Li et al. successfully utilized this method to produce uniform-sized ropivacaine microspheres with efficient drug loading and stable release by combining emulsification with a novel pre-mixed membrane technique [49] (Figure 3c). This approach has also been used to create chromatography microsphere media, such as polysaccharide composite microspheres and chitosan microparticles [50,51]. Membrane emulsification offers benefits such as a narrow particle size distribution, stable particle size, ease of operation, low energy consumption, and mild operating conditions suitable for sensitive drugs. However, the choice of membrane material is a limiting factor. Ideally, the membrane material should possess specific characteristics such as narrow pore size, low flow resistance, high mechanical strength, chemical durability, heat resistance, sterilizability, and cost-effectiveness. Most microfiltration membranes, with the exception of porous glass membranes, are unsuitable for membrane emulsification because of their wide pore size distribution, poor morphology, and insufficient solvent resistance. Therefore, the development of specialized porous membranes, particularly polymer-based ones, is crucial for advancing membrane emulsification technology.

**Figure 3 ijms-25-06556-f003:**
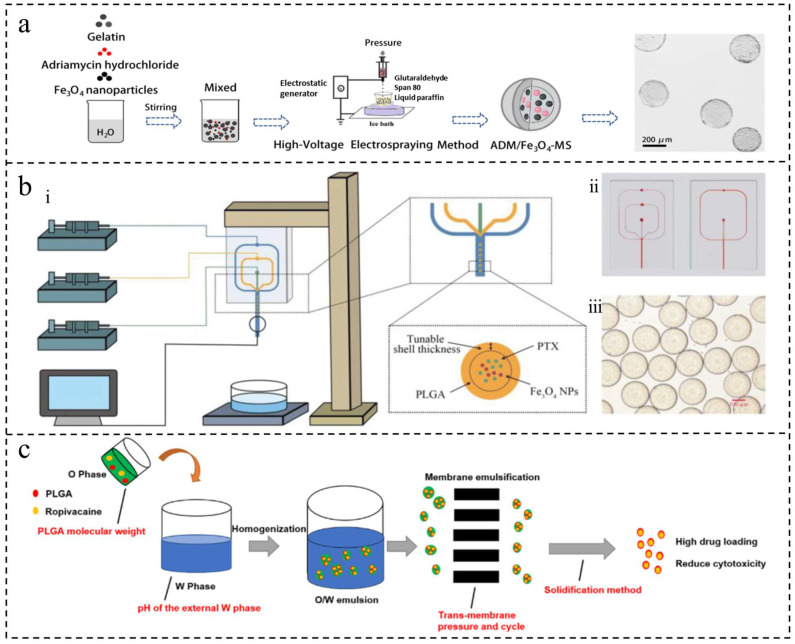
Microspheres were prepared using electrospray, microfluidic, and membrane emulsification methods. (**a**) The mechanism of preparing microspheres by the electrospray method [19]. Copyright 2022, The Author(s). (**b**) The mechanism of preparing microspheres by the microfluidic method. (i) Schematic illustration of the three-phase flow-focusing microfluidic set up. (ii) Photographs of three-phase (left) and two-phase (right) microfluidic chips. (iii) Optical photograph of PLGA microdroplets prepared via microfluidic device [47]. Copyright CCBY, 2021, Informa UK Limited (**c**) The mechanism of preparing microspheres by the membrane emulsification method [49]. Copyright 2018, Elsevier B.V.

## 3. Polymer Microspheres in Cancer Applications

This section explores the various applications of polymer microspheres in cancer diagnosis, treatment, and postoperative repair. The use of polymer microspheres in cancer treatment includes chemotherapy, immunotherapy, and radiotherapy.

### 3.1. Cancer Diagnosis and Prognostic Evaluation

Polymer microspheres have the potential to detect biomarkers associated with cancer diagnosis and prognostic evaluation, aiding healthcare professionals in early cancer detection and prognosis. In the following sections, we will provide a concise overview of how polymer microspheres are utilized in cancer diagnosis and prognostic assessment.

#### 3.1.1. Circulating Tumor Cells (CTCs)

CTCs are cells that are released from tumor tissues into the bloodstream, serving as crucial liquid biopsies for cancer diagnosis and prognosis. Detecting CTCs can provide personalized treatment options for patients and illuminate the mechanisms of cancer metastasis [52]. However, the challenge lies in the extremely low concentration of CTCs in the blood, complicating the analysis process. Currently, there are two primary methods for isolating CTCs: antibody-dependent biological isolation and non-antibody-dependent physical isolation [53]. The former is constrained by the limited knowledge of antibodies and the dynamic expression of CTC markers, while the latter may face limitations due to the similarity in size between CTCs and leukocytes, or interference from high concentrations of blood cells [54].

Yu et al. developed hollow suspended immunomagnetic microspheres (FIMMs) based on Dong et al.’s work, utilizing buoyancy force to float in the blood and magnetic field control to prevent cell damage and ensure efficient CTC capture [55,56]. With a capture rate of 93% and a detection limit of 5 cells/mL, FIMMs were successfully used to detect CTCs in blood samples from patients with epithelial cancer. On the other hand, Qiu et al. utilized a combination of polystyrene (PS) microspheres and acoustic-sensitive particles in an acoustic fluidic microarray system for CTC isolation [57]. Yin et al. integrated antibody and non-antibody approaches using silica-based cone microcavity arrays (PMCAs) and fluorescent PS microspheres to isolate CTCs from whole blood. This streamlined the immunostaining process and successfully isolated and characterized CTCs in 15 colorectal cancer patients [58] (Figure 4a).

#### 3.1.2. Tumor-Associated Antigen (TAA)

Immunoassays excel at identifying antigens and antibodies, but conventional immunosensors are limited to measuring one biomarker at a time, which restricts the accuracy of tumor diagnosis. Cancer is a complex disease that requires the simultaneous detection of multiple biomarkers for accurate prognosis. Multiplexed immunoassays enable the simultaneous measurement of multiple target molecules, utilizing technologies such as planar arrays and liquid suspension biochips. Among these, suspension-encoded microspheres combined with flow cytometry enable high-throughput detection. In a study by Tang et al., fluorescent microspheres of maleic anhydride-grafted PLA (PLA-MA) embedded with quantum dots were used for multiple immunoassays of tumor markers using SPE membrane emulsification [59]. In contrast, Li et al. developed magnetic fluorescent encoded microspheres embedded with two quantum dots, allowing for more barcodes to be distinguished [60], while Wei et al. created Janus magnetophotonic crystal microspheres using droplet microfluidics for exosome barcode analysis [61] (Figure 4b). Clinical experiments have demonstrated that these techniques significantly enhance the sensitivity of detection and efficiency of enrichment for analyzing multiple tumor markers.

#### 3.1.3. MicroRNA (miRNA)

miRNA plays a crucial role in cancer; however, the absence of standardized detection methods leads to non-comparable results. Due to their low abundance and concentration, there is a demand for accurate, cost-effective assays with high sensitivity and specificity [62]. Tania et al. utilized a molecular beacon probe integrated into the surface of a supramolecular PEG hydrogel to directly detect miR-21 in small sample volumes. This approach achieved high sensitivity without the need for prior amplification and enabled rapid analysis [63] (Figure 4c). Subsequently, they improved this probe by combining it with microgel technology, allowing for the accurate detection of the breast cancer marker miR-103-3p in real samples with better accuracy, linearity, and precision compared to Real-Time Quantitative Reverse Transcription PCR [64]. This approach not only reduces detection time but also minimizes the risk of sample contamination.

**Figure 4 ijms-25-06556-f004:**
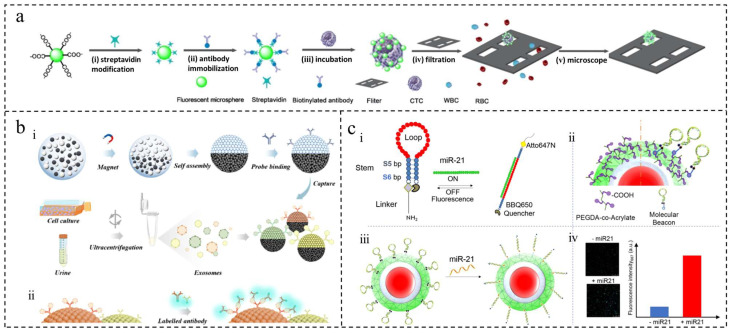
Polymer microspheres were used to detect CTCs, TAA, and miRNA. (**a**) The principle of fluorescence microsphere-mediated amplification technology for separating and analyzing CTCs. (i) Fluorescent microspheres are modified with streptavidin. (ii) Biotinylated antibodies are attached to the microspheres. (iii) The microspheres bind to CTCs in patient blood samples. (iv) The sample is filtered using a silicon-based array. (v) CTCs are counted using a fluorescence microscope [58]. Copyright 2020, American Chemical Society (**b**) Diagram of Janus magnetic microspheres for noninvasive analysis of bladder cancer-derived urinary exosomes. (i) Prepare microspheres and immobilize antibody probes for exosome capture. (ii) Multiplex exosome analysis platform using Janus magnetic microsphere barcodes [61]. Copyright© 2022, American Chemical Society (**c**) Molecular beacon-conjugated microgel design and mechanism of biosensing. (i) Two molecular beacons were designed, differing in stem oligonucleotides (S5–S6). (ii) Fluorescent microgels covalently bind to these beacons. (iii) Microgel–beacon conjugates are mixed with the samples. (iv) Fluorescence intensity measured by microscopy or spectrofluorometry [63]. Copyright 2019, American Chemical Society.

### 3.2. Treatment of Tumors

In chemotherapy, immunotherapy, and physiotherapy, polymer microspheres have brought new breakthroughs in cancer treatment with their unique advantages. This section will delve into the application of polymer microspheres in these therapies and present the latest research progress on polymer microspheres for cancer treatment in each type of therapy in detail, as illustrated in Table 2, to offer readers a comprehensive and in-depth understanding.

#### 3.2.1. Chemotherapy

Chemotherapeutic drugs play a vital role in cancer treatment, but their serious side effects can inadvertently harm healthy cells. Utilizing microspheres as drug carriers enables targeted delivery of chemotherapy drugs to the tumor site, improving therapeutic outcomes and minimizing side effects. Therefore, incorporating microspheres to transport chemotherapeutic drugs not only enhances treatment efficacy but also reduces patient discomfort, leading to the development of innovative cancer therapies.

##### Adriamycin/Doxorubicin (DOX)

DOX is a potent antitumor chemotherapeutic agent that inhibits nucleic acid synthesis. It is commonly used in clinical settings for various tumor treatments and is often encapsulated in microspheres [19,65]. Research has demonstrated that DOX-loaded magnetic microspheres can autonomously regulate thermotherapy temperature and release the drug in a controlled manner without causing toxicity [38]. These microspheres have shown promise in blocking blood vessels in a rabbit ear artery model and are detectable under Computed Tomography (CT)/Magnetic Resonance Imaging (MRI), indicating their potential use as embolic agents. Additionally, gelatin microspheres co-loaded with DOX and Fe_3_O_4_ have been utilized in MRI to enhance the efficiency and efficacy of microwave thermotherapy [19]. In the treatment of osteosarcoma, a complex of hydrogel microspheres containing collagenase, PLGA microspheres with pioglitazone, and DOX has been reported to inhibit tumor growth by degrading the tumor extracellular matrix using collagenase [27]. The fluorescence of DOX enables the detection of adriamycin and the destruction of tumor vascular endothelial cells with 5,6-dimethylflavone-4-acetic acid (DMXAA) loaded onto mesoporous silica-coated gold nano-rods (AuRN@SiO_2_) and photocrosslinked gelatin (GelMA) microgels, allowing for multiple rounds of light-controlled therapy with reduced drug dosage and improved anti-capillary efficiency [20]. The use of microsphere-loaded DOX has been shown to reduce drug resistance and cardiotoxicity, enhance drug loading efficiency and release time [66,67] (Figure 5a,b), and improve therapeutic efficacy when co-delivered with other drugs [68,69]. Therefore, the application of microsphere-loaded adriamycin holds promise in cancer therapy.

##### Pentafluorouracil (5-FU)

5-FU, a well-known antitumor drug that disrupts nucleic acid synthesis, was studied for its encapsulation in lysozyme microspheres and AuNR through acoustic chemistry [70]. Drug release was facilitated by ultrasound, leading to its transfer into cancer cells via temporary pores (acoustic pores) on the cell surface, effectively killing the cells and demonstrating potent anticancer activity. In a separate investigation on oral drug delivery, DOX and 5-FU were co-loaded onto layered double hydroxide, encapsulated with carboxymethyl starch (CMS), resulting in 72 μm microspheres [69]. The release rates of the two drugs were 22% and 29%, respectively, showcasing sustained release effects on colon cancer cells and suggesting a promising formulation for oral co-administration. Furthermore, research has explored a combined system of gelatin microspheres and an alginate methacrylate (Alg-MA) hydrogel for the targeted delivery of 5-FU to the stomach, opening up new possibilities for the application of this drug [71].

##### Natural Products

Natural products are a valuable resource for the development of new drugs due to their potent biological activity, multi-targeted effects, and minimal side effects. For instance, artesunate (ART), derived from artemisia annua in traditional Chinese medicine, has shown efficacy in treating non-small-cell lung cancer [28]. Research has demonstrated that microspheres loaded with ART and PLGA can steadily release the compound, effectively inhibiting the proliferation, migration, and invasion of lung adenocarcinoma cells. Additionally, studies have highlighted the use of gelatin hydrogel particles containing curcumin, prepared through microfluidic electrospray technology, which exhibited strong toxicity against gastric cancer cells while sparing normal cells [42]. Lignans, found in herbal botanicals, have shown therapeutic benefits against malignant tumors, with microspheres serving as carriers to enhance solubility and bioavailability [72]. Similarly, the therapeutic potential of celastrol from radix rehmanniae is constrained by its low water solubility, bioavailability, and narrow therapeutic range; however, utilizing microspheres as carriers can amplify its therapeutic effects and mitigate toxicity [73].

##### Platinum-Based Drugs

Platinum-based drugs are commonly used in chemotherapy to treat tumors by damaging tumor cell DNA, inhibiting replication, and inducing apoptosis. To enhance effectiveness, platinum drugs are often combined with other anti-tumor medications. For instance, researchers developed paclitaxel-loaded PLA microspheres co-encapsulated with cisplatin in a heat-sensitive hydrogel for localized chemotherapy [74]. This temperature-sensitive system allows sustained drug release, and in vitro studies validated its ability to induce apoptosis and inhibit cell migration. In vivo experiments further confirmed its significant tumor growth inhibition.

**Table 2 ijms-25-06556-t002:** Research progress of some polymer microspheres in cancer chemotherapy, immunotherapy, and physiotherapy.

Treatment	Polymer Material	Type of Cancer	References
Chemotherapy	Carboxymethyl cellulose	Breast cancer	[66]
	Carboxymethyl starch	Colorectal cancer	[69]
	Gelatin	Osteosarcoma	[20]
	Gelatin	Hepatocellular carcinoma	[19]
	Gelatin/methacrylated alginate hydrogel composite	Gastric cancer	[71]
	Hyaluronic acid	Non-small-cell lung cancer	[5]
	Methacrylate fish gelatin	Gastric cancer	[42]
	Methacrylate gelatin	Melanoma/Breast cancer	[75]
	PLA	Melanoma	[74]
	PLGA	Lung cancer	[68]
	PLGA	Osteosarcoma	[27]
	PLGA	Malignant pleural mesothelioma	[73]
	PLGA	Non-small-cell lung cancer	[28]
	Polybutylene adipate co-terephthalate/polyvinylpyrrolidone composite	Non-small-cell lung cancer	[76]
	Porous hydroxyapatite/gelatin composite	Osteosarcoma	[67]
Immunotherapy	Chitosan/sodium alginate composite	Colorectal cancer	[77]
	Gelatin	Ovary carcinoma	[78]
	Hyaluronic acid	Pancreatic ductal adencarcinoma	[79]
	Methacrylate gelatin/methacrylate hyaluronic acid composite	Breast cancer	[80]
	PLGA	Breast cancer/melanoma	[81]
	PLGA	Melanoma	[82]
	Poly(ethylene glycol)diacrylate	Non-small-cell lung cancer	[83]
	Poly(N-isopropylacrylamide)	Melanoma	[84]
	Polyanhydride copolymer	Head and neck squamous cell carcinoma	[85]
	Silk fibroin	Melanoma	[21]
Physiotherapy	Alginate	Hepatocellular carcinoma	[18]
	Calcium alginate	Hepatocellular carcinoma	[86]
	Chitosan	Hepatocellular carcinoma	[39]
	Diallyl isophthalate	Liver cancer	[87]
	Gelatin	Osteosarcoma	[20]
	Gelatin	Hepatocellular carcinoma	[19]
	Gelatin, genipin, and sodium alginate composite	Hepatocellular carcinoma	[38]
	Liquid metals	Hepatocellular carcinoma	[88]
	Methacrylated gelatin	Hepatocellular carcinoma	[48]
	PLGA	Breast cancer	[89]
	Silicon dioxide	Hepatocellular carcinoma	[90]
	Silk fibroin	Hepatocellular carcinoma	[30,91]

Researchers proposed a novel approach involving Pt(IV) prodrug-initiated hydrogel particles combined with indocyanine green (ICG) for synergistic chemotherapy and phototherapy [75]. Here, Pt(IV) triggers nitrogen radical generation for photopolymerization and, upon reduction, forms highly cytotoxic Pt(II) for chemotherapy. Simultaneously, ICG contributes to photothermal heating and reactive oxygen species production, enhancing the phototherapy effect. Experimental findings demonstrated the efficacy of these particles in vitro for cancer cell destruction and in vivo for substantial tumor growth inhibition.

##### Other Chemotherapy Drugs

In a study on liver cancer, researchers combined embolization with hypoxia-targeted therapy using CalliSpheres microspheres (CSMTPZs) loaded with tirapazine (TPZ) [92]. TPZ, a hypoxia-selective drug, synergistically enhanced the delivery of TPZ in the tumor when combined with embolization, resulting in improved therapeutic efficacy. This approach demonstrated increased hypoxic cytotoxicity, tumor apoptosis, and necrosis while maintaining safety. Another study focused on preparing hollow microspheres of polybutylene adipate/polyvinylpyrrolidone (PBAT/PVP) loaded with erlotinib, an epidermal growth factor receptor complex kinase inhibitor used in non-small-cell lung cancer treatment, providing a foundation for lung cancer therapy [76]. Additionally, a study on erlotinib explored the design of pH-modulated hydrogel encapsulation for sustained local delivery, achieving high tumor suppression efficiency with minimal toxicity to healthy organs [5]. In the realm of preoperative chemotherapy for colon cancer, researchers developed capecitabine (CAP) and oxitinib (OSI) combined with colon-targeted particles (COPMP) for sustained drug release. This combination exhibited a strong synergistic effect, resulting in a high tumor inhibitory rate of 94%, offering a novel approach to preoperative chemotherapy for colon cancer [93].

#### 3.2.2. Immunotherapy

Immunotherapy aims to treat cancer by boosting the immune system to attack cancer cells. However, its success is limited by tumor conditions, drug stability, and distribution, which can cause negative reactions, treatment failure, and cancer return. Microspheres, which carry drugs, deliver immunotherapy drugs directly to the tumor, boosting effectiveness, stabilizing drugs, and reducing side effects. Therefore, using microsphere-loaded immunotherapy drugs is beneficial for cancer treatment and could lead to new strategies for patients.

##### Cytokines

Cytokines are small molecules secreted by various cells, primarily immune cells, with functions that include regulating cell growth, differentiation, and immune response. Certain cytokines, such as interferon-gamma (IFNγ) and interleukin-2 (IL-2), can activate immune cells and inhibit tumor growth. In the field of tumor immunotherapy, cytokines have been utilized in clinical settings. Recombinant interferon-alpha (IFNα) therapy was approved for hairy cell leukemia in 1986 [94], followed by high-dose IL-2 in 1992 for metastatic renal cell carcinoma [85]. However, the use of cytokines has been limited due to dose-limiting toxicities, such as cytokine storm [95]. Therefore, the development of drug-delivery platforms to administer cytokines to patients over extended periods of time, aiming for effective treatment while minimizing high-dose side effects, is a promising approach.

Polymeric particles (MPs) were synthesized using FDA-approved polyanhydride to encapsulate interleukin-1α (IL-1α) in a study [85]. These IL-1α-loaded MPs released IL-1α at a slower and more sustained rate compared to free IL-1α, thereby reducing associated toxicity. Another study involved preparing hydrogel microspheres conjugated with interleukin-15 (IL-15), which, when administered subcutaneously in mice for an antitumor assay, showed that a single injection significantly prolonged the half-life of immune cells and exhibited high anticancer activity [95]. Furthermore, adjusting the levels of immunosuppressive factors is also considered a viable method for tumor immunotherapy. For example, a polymeric carrier made of sodium alginate combined with chitosan encapsulating a transforming growth factor-β (TGF-β) receptor inhibitor and E. coli induced apoptosis of tumor cells and differentiation of immune cells in the intestine, thereby enhancing the immunotherapeutic effect [77]. Selective depletion of immunosuppressive cytokines, such as using heparin-coupled PVA microspheres to remove vascular endothelial growth factor (VEGF) and TGF-β, has shown significant antiproliferative and pro-apoptotic effects on breast cancer cells [29]. Hence, selective depletion of immunosuppressive cytokines holds promise as a strategy for cancer treatment.

##### Antibodies

Antibodies, also known as immunoglobulins, are produced by the body’s immune system in response to antigenic stimuli. Their primary function is to recognize and bind specifically to foreign substances such as bacteria or viruses. In cancer immunotherapy, antibodies are crucial, with approaches including monoclonal antibody therapy, antibody-drug conjugate (ADC) therapy, and bispecific antibody therapy [96]. Monoclonal antibody drugs are commonly used in treating various tumors, but face challenges such as low stability, short half-life, and limited bioavailability, necessitating mainly intravenous administration. There is a need for alternative delivery methods in clinical practice. Moreover, combining antibody immunotherapy with other treatments holds promise for enhancing the effectiveness of tumor therapy.

Biodegradable microgel carriers offer a promising method for delivering therapeutic drugs directly to tumor sites with reduced side effects compared to intravenous administration. However, the encapsulation of monoclonal antibodies can lead to conformational changes and aggregation, impacting their effectiveness. A new microgel platform was developed using emulsification to encapsulate monoclonal antibodies through post-loading technology. This approach utilizes a superhydrophilic amphiphilic ionic polymer to stabilize the antibodies, preventing conformational changes and aggregation and enabling a controlled release of the antibody [97]. An alternative hydrogel platform involves encapsulating antibodies and cytokines in microspheres, which can enhance lymphocyte function, suppress tumor growth through local injection, and improve anti-tumor effects [81]. Moreover, the combination of immunotherapy with other treatments, such as dual-functionalized microgels containing antibodies and anti-tumor drugs to target and eliminate cancer cells selectively, shows significant promise in the fields of biomedicine and cancer therapy [83].

##### Acquired Cell Therapy (ACT)

Adoptive cell therapy (ACT) is a treatment method that involves harvesting immune cells from a patient, culturing them in vitro, and then infusing them back into the patient to target and kill tumor cells. The immune cells are typically sourced from the patient’s blood or tumor tissues and can be expanded or genetically modified to boost their anti-cancer capabilities. Examples of ACT therapies include chimeric antigen receptor T-cell immunotherapy (CAR-T), T-cell receptor therapy (TCR-T), and tumor-infiltrating lymphocyte (TIL) therapy [98]. Despite its promise, ACT faces challenges in solid tumor treatment due to the immunosuppressive tumor environment, limited proliferation of immune cells in vivo, and the risk of cytokine storm syndrome from high doses of immune cells attacking cancer cells [78].

The use of artificial antigen-presenting cells (aAPCs) made from poly(N-isopropylacrylamide) (PNIPAM) microspheres has been developed to stimulate and expand T cells in vitro, reducing the need for natural antigen-presenting cells (APCs) and thus lowering costs and time. Studies have shown that overdose immunotherapy in hormonal mice using these aAPCs successfully inhibited tumor growth [84] (Figure 5c,d). Additionally, the in vivo release of immune cells can be addressed by biomaterial delivery platforms such as gelatin microhydrogels, which locally release contents and support immune cell proliferation [78]. Encapsulating CAR-T cells in microgels has shown effective targeting and cytotoxicity in vitro and on three-dimensional tumor spheroids, potentially enhancing CAR-T immunotherapy.

##### Vaccines

Tumor vaccines have become a focal point of research in tumor therapy, aiming to control or eliminate tumors by introducing tumor antigens into the patient’s body to enhance the immune response. Despite the existence of various types of vaccines, direct use is challenging due to the rapid clearance of tumor antigens in the body and their low immunogenicity [98]. Chemical modification for antigen delivery is not efficient, hence the potential for improving vaccine efficacy through the delivery of antigens using micro- and nano-materials.

Tumor vaccine carrier selection is a complex process, as inorganic materials are slow to degrade and polymeric materials may lead to inflammatory reactions [21]. Biocompatible and degradable materials, such as hyaluronic acid [79], gelatin [80], and silk proteins from domestic silkworms [21], are considered suitable for vaccine carriers. These microsphere vaccines have shown promise in reversing tumor immunosuppression and enhancing survival rates in a safe and efficient manner. In addition to traditional vaccine delivery methods, microneedles have emerged as a potential way to administer vaccine particles through the skin, which is rich in immune cells. Zaman et al. conducted a study evaluating the effectiveness of microneedle-embedded particulate-based whole-cell lysate vaccines for treating breast cancer in a mouse tumor model [99]. Another study focused on fine-tuning microneedles to predict the release of phages by optimizing the components of microneedle patches for intradermal delivery [100]. Photothermal therapy has shown promise in inducing long-term immune memory and promoting memory T-cell differentiation, similar to in situ vaccines. One study utilized PLGA microspheres containing tumor antigens, metformin, and hollow gold nanospheres for drug release through photothermal therapy to stimulate T-cell production and enhance T-cell survival [101]. Another study explored the use of glycolysis inhibitors in photothermal therapy to reduce the tumor’s heat stress response, improve therapy efficacy, and generate more memory T cells [82].

**Figure 5 ijms-25-06556-f005:**
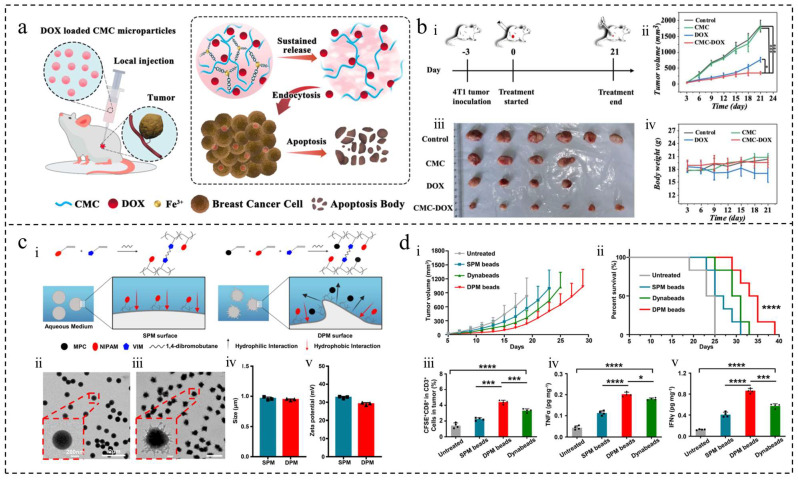
Polymer microspheres were used in tumor chemotherapy and immunotherapy. (**a**) Schematic diagram of DOX-loaded CMC microparticles for tumor injection therapy. (**b**) Antitumor study of microparticles. (i) Breast cancer model schematic. (ii) Tumor images from different mouse groups. (iii) Tumor volume changes in mice. (iv) Body weight analysis of treated mice [66]. Copyright 2022, Elsevier B.V. (**c**) Synthesis and characterization of DPM and SPM using hydrophilic monomers for morphology control. (i) Synthesis and mechanism of SPM and DPM. (ii) TEM image of SPM. (iii) TEM image of DPM. (iv) SPM and DPM sizes measured by DLS. (v) Zeta potential of SPM and DPM. (**d**) Antitumor study of DPM and SPM. (i) Tumor growth curves. (ii) Survival rates of B16-OVA tumor-bearing mice with different treatments. (iii) Percentage of CD8^+^ OT-1 T cells in B16-OVA tumor tissue by treatment. (iv) TNF-α levels in tumor tissue by treatment. (v) IFN-γ levels in tumor tissue by treatment. * *p* < 0.05, *** *p* < 0.001, **** *p* < 0.0001 [84]. Copyright 2023, Elsevier Ltd.

#### 3.2.3. Physiotherapy

While physical therapy has advanced in cancer treatment, traditional methods such as radiotherapy and thermotherapy can have significant side effects and limited effectiveness against certain cancers. Microsphere-loaded drug therapies offer new promise in cancer treatment, providing high efficiency, precision, and minimal side effects. The technology for preparing these therapies is also continuously improving. As a result, microsphere-loaded drug therapy offers more advantages than physical therapy and is anticipated to offer safer and more effective treatment options for cancer patients.

##### Radiotherapy

Brachytherapy is a crucial treatment method for advanced hepatocellular carcinoma, encompassing radioactive seed implantation (RSI) and transarterial radiation therapy embolization (TARE) [18]. TARE disrupts the nutrient supply to cancer cells by obstructing the blood vessels feeding the tumor, making it a favorable target for this therapy due to the liver’s dual blood supply [1]. However, RSI and TARE alone are insufficient for eradicating tumor metastasis and recurrence, limiting their clinical utility. In recent years, various strategies involving microsphere-encapsulated radionuclides have enhanced the applicability of TAE. It is essential to consider factors such as nuclide selection (whether it is multifunctional), labeling strategy (ensuring safety), microsphere material selection (degradability and deformability), particle size uniformity, and the incorporation of combination therapy strategies when utilizing this approach.

In terms of nuclide selection, ^90^Y-based microspheres such as TheraSphere and SIR-Spheres, approved by the FDA for hepatocellular carcinoma treatment, are pure beta-emitters and are not suitable for nuclear imaging or post-treatment monitoring [90]. Recent advancements include microspheres loaded with ^111^In, ^131^I, ^166^Ho, and ^177^Lu, capable of therapeutic and imaging functions [39]. Regarding nuclide labeling, strict regulation is essential for safety. Biomaterials such as polydopamine (PDA), sodium alginate, filipin proteins, and GelMA enhance nuclide labeling stability. PDA improves radionuclide chelation efficiency and stability in organisms. Filipin proteins and sericin proteins rich in tyrosine have been used for chelating ^177^Lu and ^131^I, respectively [30,91]. GelMA, with tyrosine-rich residues, was successfully grafted onto microspheres, producing ^131^I-labeled GelMA radiolabeled microspheres (^131^I-GMs), enhancing radionuclide stability and reducing leakage [48].

In the context of microsphere material selection, microspheres can be categorized as degradable or non-degradable based on their degradation properties [91]. Biodegradable microspheres, upon degradation, promote vascular recanalization, offering a pathway for consistent tumor embolization with minimal damage to the organism, even in cases of rare ectopic embolization. As a result, there is a growing focus on the research and development of biodegradable materials. Chitosan microspheres [39], sodium alginate microspheres [18], filipin protein microspheres [30,91], and CelMA microspheres [48] are among the widely utilized options. Apart from biodegradability and good biocompatibility, which are necessary for microsphere re-embolization, uniform and controllable size, as well as good elasticity, are also crucial characteristics for radioactive microspheres [48]. Microspheres with uneven sizes tend to accumulate in proximal vessels, impacting the distal therapeutic outcomes, while insufficient elasticity can hinder smooth injection due to catheter blockage. Microfluidic technology enables the production of microspheres with uniform size distribution and optimal elasticity, making it suitable for radioactive microsphere preparation [48,75,88,89]. Furthermore, in terms of combined treatment approaches, microspheres loaded with a single radionuclide typically do not lead to complete tumor eradication, and liver cancer metastasis and recurrence remain prevalent. Therefore, combining radionuclide therapy with other therapeutic modalities such as chemotherapy, immunotherapy, photothermal therapy, and photodynamic therapy is crucial to enhance the efficacy of cancer treatment [18,87].

##### Magnetothermal Therapy (MHT)

Magnetic hyperthermia therapy (MHT) is a form of physical therapy that leverages the use of alternating magnetic fields (AMFs) to induce thermal energy in magnetically sensitive materials, ultimately leading to the apoptosis or necrosis of tumor cells. Additionally, these magnetically sensitive materials can also serve an imaging purpose in CT, X-ray, and MRI scans, allowing for precise positioning of microspheres during treatment monitoring and postoperative evaluations [38,102]. By loading microspheres with magnetically sensitive materials, a dual therapeutic and imaging function can be achieved. Commonly used magnetically sensitive materials include Fe_3_O_4_ nanoparticles (NPs), liquid metals (LMs), and superparamagnetic iron oxide nanoparticles (SPIONs) [38,86,87,88,102].

Wei et al. developed magnetic microspheres using low-temperature superparamagnetic iron oxide NPs [38] (Figure 6a,b), while Xiao et al. and Chen et al. loaded microspheres with nano-Fe_3_O_4_ for MHT in combination with TRAE treatment, CT, or MR imaging [87,102]. Liquid metals (LMs) are a promising class of flexible functional materials with properties such as good biocompatibility, flowability, and thermal/electrical conductivity, making them attractive for various biomedical applications, especially in cancer therapy [86,88]. LMs exhibit a more pronounced vortex thermal effect under AMF compared to magnetic NPs. Yang et al. utilized microfluidic technology to create homogeneous liquid metal microspheres for effective cancer treatment through MHT combined with TAE [88].

##### Phototherapy

Photodynamic therapy (PDT) and photothermal therapy (PTT) have garnered significant interest in recent years for their advantages, including being less invasive, less toxic, more efficacious, more selective, more reproducible, and with fewer side effects [103]. In the study of these therapies, key materials such as photothermal-sensitive gold NPs, layered 2D nano-materials, and liquid metals are commonly utilized. MXene nanosheets are particularly favored for their large surface area, strong near-infrared absorbance, and good biocompatibility.

Gao et al. successfully fabricated three-dimensional CNT/MXene microspheres with rough surfaces and large surface areas using the template and spray drying methods [103]. The incorporation of carbon nanotubes (CNTs) notably expanded the absorption spectrum of TiO_2_ on the nanosheet surface into the visible region. The microspheres demonstrate unique photothermal effects and stability when exposed to near-infrared laser irradiation, efficiently generating single-line oxygen under 650 and 808 nm light. Additionally, magnetic liquid metal nanoparticles (Fe@EGaIn NPs) exhibit dual-mode imaging capabilities in CT and MR, along with photothermal/photodynamic sensitization properties for photothermal conversion and ROS generation. Wang et al. developed Fe@EGaIn/CA microspheres, integrating the imaging and photothermal/photodynamic functions of magnetic liquid metal nanoparticles ( Fe @ EGaIn NP ) and calcium alginate (CA) microspheres, offering new possibilities for cancer therapy through embolization, drug-filling, and enhanced imaging [86]. On another front, AuNRs with tunable local surface plasmon resonance (SPR) are crucial for the advancement of near-infrared (NIR)-responsive cellular imaging and therapeutic platforms. Yan et al. successfully immobilized mesoporous silica-coated AuNR@SiO_2_ core-shell nanoparticles in monodispersed GelMA microgels using microfluidics for the chemo-photothermal treatment of solid tumors [20]. In contrast, Wang et al. developed materials with ultra-broad absorption in the visible and near-infrared regions by synthesizing and encapsulating AuNPs in a gelatin microgel network, while incorporating photosensitizers to enable a novel strategy for combined PTT and PDT cancer therapy [104].

##### Other Physiotherapy

Apart from physiotherapy, various other physical therapies are utilized in cancer treatment. Thermotherapy, for example, can enhance tumor sensitivity to chemotherapeutic agents, improve drug delivery efficiency, and alter the tumor microenvironment [30,91]. Chen et al. developed gelatin microspheres loaded with adriamycin and Fe_3_O_4_NPs, allowing for visual monitoring under MRI and enhancing the killing effect of chemotherapy through microwave thermotherapy [19]. Irreversible electroporation (IRE) is a promising non-thermal ablative therapy for cancer treatment. Liu et al. demonstrated that an injectable hydrogel vaccine combined with ablative therapy promoted anti-tumor immunity in a mouse model of pancreatic cancer [79]. Overall, these physical therapies offer new treatment options and hope for cancer patients by enhancing therapeutic effects through different mechanisms.

### 3.3. Postoperative Tissue Repair and Regeneration

The primary focus of postoperative tumor therapy is to prevent cancer cell recurrence and metastasis, while also promoting tissue regeneration at the surgical site. However, current therapeutic materials have shown limited effectiveness in this regard, prompting researchers to explore the development of innovative multifunctional therapeutic systems. To this end, these systems aim to achieve the dual objectives of treatment and reconstruction following tumor surgery [105].

A study successfully prepared a novel microgel using microfluidic electrospray and liquid nitrogen technology, capable of carrying cryo-shocked cancer cells, immune adjuvants, and immune checkpoint inhibitors [80] (Figure 6c,d). The microgel demonstrated good biocompatibility and degradation properties. Experimental results in a mouse model of breast cancer revealed that the microgel not only supported tissue regeneration but also effectively suppressed cancer cell recurrence and metastasis, offering new possibilities for post-operative tumor treatment and tissue repair. Another study introduced an injectable macroporous home silkworm sericin microsphere, utilizing sericin protein as a matrix [21]. These microspheres can inhibit tumors and minimize surgical damage through a single injection. Their macroporous structure not only facilitates immune cell aggregation but also promotes healthy tissue regeneration, effectively eliminating the tumor and healing the wound site simultaneously. Furthermore, in gastric cancer research, Zhu et al. developed CelMA hydrogel particles that encapsulate curcumin using microfluidic electrospray technology [42]. These microspheres exhibited biocompatibility and demonstrated significant tissue regeneration effects in in vivo experiments. Collectively, these studies highlight the potential of novel biocompatible materials for postoperative tumor therapy and tissue regeneration, offering new strategies and approaches for future tumor therapy. Beyond tumor therapy, these materials also play a crucial role in cell culture, wound healing, and tissue engineering.

## 4. Conclusions and Outlook

As mentioned earlier, polymer microspheres are derived from a wide range of feedstocks, including both natural and synthetic polymers, with the selection of feedstocks tailored to the intended application to impart specific properties to the polymer microspheres. Various preparation methods exist, each offering unique characteristics and options for microsphere production, allowing for flexibility based on application needs. This article also explores the use of polymer microspheres in cancer diagnosis, therapy, and postoperative repair, focusing particularly on their current role in cancer treatment. In conclusion, polymer microspheres serve as an innovative drug delivery system with significant implications for cancer research. The abundance of raw materials, diverse preparation methods, and promising applications position polymer microspheres as potential game-changers in cancer therapy moving forward.

In recent decades, advancements in microsphere preparation technology and the introduction of new materials have significantly advanced the use of polymer microspheres in cancer research. The industrialization of cancer-related polymer microsphere products has had a positive impact on global healthcare, enhancing people’s quality of life. In experimental research, the creation of multifunctional and innovative polymer microspheres has played a crucial role in advancing the field. However, the rapid growth of polymer microspheres has brought about certain challenges, particularly in the areas of large-scale production, material safety, and expanding application possibilities. In the following paragraphs, these problems are introduced and the direction of future development is discussed.
The scale of production, mass production, and industrialization of microspheres are crucial for their widespread clinical application. For instance, both microfluidic technology and membrane emulsification technology offer advantages in producing uniform microsphere products, but they also present challenges to industrialization. While microfluidic technology can create uniform microspheres, issues such as equipment cleaning difficulties and low production efficiency hinder its industrial application. On the other hand, membrane emulsification technology can regulate product uniformity through membrane pore size, but it demands high-quality membranes in production equipment and faces challenges in mass production. Therefore, future research should prioritize the use of high-throughput equipment and technologies for microsphere preparation, reducing technological barriers and equipment costs and enhancing the feasibility of industrial mass production of microspheres.The urgent need to address the in vivo behavior and safety of microspheres is evident. While laboratory research is advancing with the development of new materials, only a limited number of polymer materials have been approved for clinical in vivo use. Furthermore, the preparation of certain polymers may necessitate the use of additional initiators, metal ions, or other cross-linking agents, which could potentially have adverse effects on patients. To tackle this challenge, it is imperative to focus on the development of more biomaterials with superior biocompatibility and to enhance research on in vivo safety experiments.The application fields of microspheres have been expanding in recent years. Apart from tumor therapy, they also play a significant role in cell culture, wound healing, and tissue engineering. Microspheres serve as carriers in cell culture to enhance the cell growth environment and stimulate proliferation and differentiation. During the wound-healing process, microspheres can transport growth factors and other active substances to speed up healing and repair. In tissue engineering, microspheres act as scaffolding materials to direct cell growth and tissue regeneration, offering valuable support for the advancement of regenerative medicine.

In summary, the development of microspheres in cancer-related fields shows promise, but challenges such as large-scale production, material safety, and the expansion of application fields need to be addressed. Future research should concentrate on enhancing the efficiency of microsphere preparation, exploring new biomaterials, and broadening the application areas of microspheres to advance the technology’s development and utilization.

## Figures and Tables

**Figure 1 ijms-25-06556-f001:**
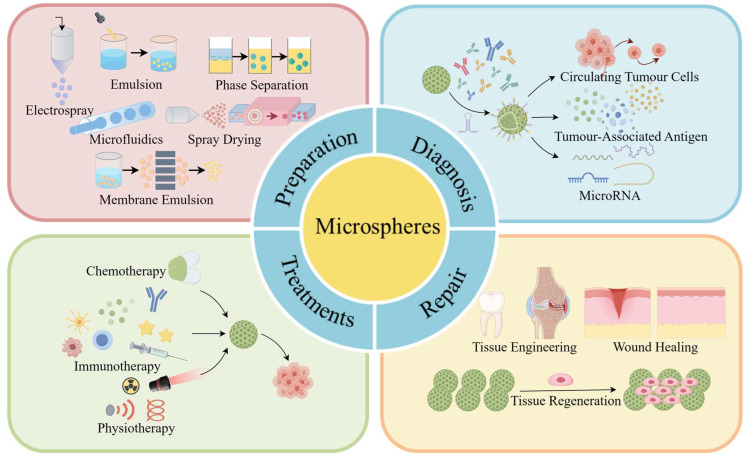
A diagram of polymer microspheres showing their preparation and application in cancer. This schematic diagram comprehensively illustrates various preparation methods of polymer microspheres, as well as the utilization of modified polymer microspheres in detecting circulating tumor cells, tumor-associated antigens, and microRNAs. It also reveals how polymer microspheres can participate in cancer treatment through chemotherapy, immunotherapy, and physiotherapy while playing a critical role as an effective carrier in tissue repair (Drawn by Figdraw 2.0).

**Figure 6 ijms-25-06556-f006:**
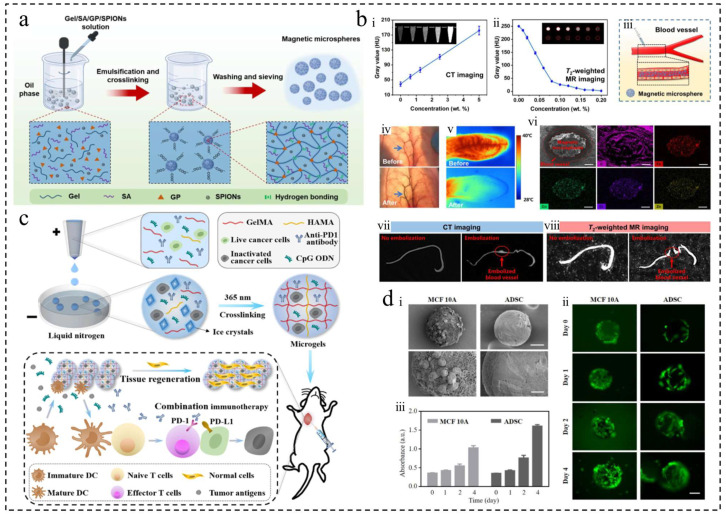
Polymer microspheres were used for physical therapy and postoperative repair of tumors. (**a**) Preparation of magnetic microspheres. (**b**) CT/MR imaging and in vivo embolization study. (i) In vitro CT. (ii) T2-weighted MR of magnetic microspheres. (iii) Embolized blood vessel schematic. (iv) Rabbit auricular artery photos before/after embolization. The blue arrow indicates the rabbit ear artery. (v) Rabbit ear IR thermal images before/after. (vi) Cross-section morphology and element analysis of embolized vessels. The estimated scale bar is 200 μm. (vii) In vivo CT. (viii) T2-weighted MR of rabbit ears in transverse section [38]. Copyright 2022, Elsevier Ltd. (**c**) Schematic of cryo-shocked cancer cell microgels for postoperative immunotherapy and tissue regeneration. (**d**) Microgel’s in vitro tissue reconstruction: (i) SEM images of blank microgels co-cultured with MCF 10A or ADSCs for 24 h; 50 μm and 20 μm scale bars. (ii) Fluorescence images depict Calcein AM-stained cells adhered to microgels, exhibiting a green color; 100 μm scale bar. (iii) Cell proliferation analysis of adhered cells by CCK8 assay (*n* = 4); mean ± SD [80]. Copyright CCBY, 2023, Elsevier B.V.

**Table 1 ijms-25-06556-t001:** The advantages and disadvantages of microsphere preparation methods.

Method	Advantages	Disadvantages	References
Emulsification	1. Simple operation and low equipment cost	1. Wide particle size distribution 2. Use of one or more surfactants 3. Low yield	[5,30,31]
Phase separation	1. Simple equipment	1. Microspheres are easy to agglomerate and difficult to separate	[32]
Spray drying	1. High productivity without additional drying process; 2. Suitable for a wide range of drugs 3. High encapsulation rate	1. Part of the undried raw material adheres to the inner wall of the instrument, resulting in loss of material. 2. Precise control of the drying temperature is required	[6,33]
Electrospray	1. Adjustable microsphere size and shape 2. High encapsulation rate 3. Preparation of complex structures	1. High equipment cost and complex operation 2. Requirements for material properties 3. Slow preparation process and low yield 4. Highly volatile solvents present safety hazards and pollution problems	[19,34]
Microfluidics	1. Adjustable microsphere size and shape 2. Suitable for a variety of materials and reaction conditions 3. Small amount of reagents and low cost	1. Complexity of operation and need for precise control 2. Unsuitable for large-scale production 3. High and time-consuming equipment maintenance	[35,36]
Membrane emulsification	1. Narrow particle size distribution of droplets 2. Easy operation and low energy consumption 3. Gentle operating conditions	1. High equipment requirements, most microfiltration membranes are not suitable for membrane emulsification	[37]

## Data Availability

Not applicable.

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
