# Peer review of "Polymer Microspheres and Their Application in Cancer Diagnosis and Treatment"

_ijms, 2024, doi:10.3390/ijms25126556_

Round 1

Reviewer 1 Report

Comments and Suggestions for Authors

The review article titled "Biomedical microspheres and their application in cancer diagnosis and treatment" provides a comprehensive description of commonly used microsphere materials, summarizes various methods of their preparation, and describes applications these microspheres in the diagnosis, therapy and post-operative care of cancer. The structure of the work is very good and the content is presented in a logical and understandable way. The work constitutes a significant contribution to systematizing the current state of knowledge regarding the use of microspheres in the diagnosis and treatment of cancer. It is also worth emphasizing that there are 106 references, mostly publications from the last 5 years.

The work should be published, but it would be good to make small changes.

1. English editing should be made to get rid of minor linguistic errors.

2. A description of Figure 1 should be added.

3. It would be good to summarize, for example, in a table which microspheres (made of what material) are used in which treatment (chemotherapy, immunotherapy, etc.) and in which cancer it was confirmed.

Comments on the Quality of English Language

English editing should be made to get rid of minor linguistic errors.

Reviewer 2 Report

Comments and Suggestions for Authors

I am unable to recommend publication of this review paper because its topic is vague. Microspheres, unlike liposomes or other established carrriers, is a term describing the morphology of the vehicle. In this review, the authors selected very limited nunber of works covering spherical particles. They do not cover whole classes of inorganic drug carriers, as calcium carbonate spheres, or organic, as layer-by-layer capsules, etc. Essentially, the carriers summarized here are selected only based on their shape, while the chemistry and function are not covered well. As such, this review does not contribute to the field

Comments on the Quality of English Language

N/a

Reviewer 3 Report

Comments and Suggestions for Authors

Microspheres have gained popularity due to their rapid and simple preparation process, excellent performance in drug release, and good compatibility with biological systems. This review paper focused on the most used materials for microsphere preparation and examined how microspheres were currently utilized in areas such as cancer diagnosis and therapy. The examples used are very thorough and convincing, with schematics well presented. In all, this review is well organized, and I would recommend the publication if the authors could address the following minor issues:

1.      In table 1, the advantages of double mulsification include more homogeneous microspheres, while the disadvantages include wide particle size distribution. This is confusing and should be restated.

2.      In Figure 2 c, the spray drying method, what do these SEM images represent? Are they in different phases of the same preparation or different microspheres? Please comment on this in the caption. 

Round 2

Reviewer 2 Report

Comments and Suggestions for Authors

1. Regarding this question: Are the references cited in this manuscript appropriate and relevant to this research?

- I did not check the relevance of the references cited. This is out of the scope of a referee to check all >100 references for relevance. I assume that the authors should do this.

2. I can only repeat my previous suggestion: "I am unable to recommend publication of this review paper because its topic is vague. Microspheres, unlike liposomes or other established carrriers, is a term describing the morphology of the vehicle. In this review, the authors selected very limited nunber of works covering spherical particles. They do not cover whole classes of inorganic drug carriers, as calcium carbonate spheres, or organic, as layer-by-layer capsules, etc. Essentially, the carriers summarized here are selected only based on their shape, while the chemistry and function are not covered well. As such, this review does not contribute to the field"

Perhaps, the authors should consider re-writing the paper based on the chemistry AND morphology (e.g. polymer microspheres)
